# COVID-19 and Gastrointestinal Symptoms—A Case Report

**DOI:** 10.3390/geriatrics5020031

**Published:** 2020-05-15

**Authors:** Alistair J. Mackett, Victoria L. Keevil

**Affiliations:** 1Department of Medicine for the Elderly, Cambridge University Hospitals NHS Foundation Trust, Cambridge CB2 0QQ, UK; alistair.mackett@addenbrookes.nhs.uk; 2Department of Medicine, University of Cambridge, Cambridge CB2 0QQ, UK; 3Cambridge Institute of Public Health, University of Cambridge, Cambridge CB2 0SR, UK

**Keywords:** COVID-19, older, gastrointestinal

## Abstract

COVID-19, a new illness secondary to a novel Coronavirus emerged in December 2019 in China. Our early understanding of the clinical features of COVID-19 has been based on case series emerging from the first outbreak in Wuhan. These features included fever, a dry cough, myalgia and dyspnea. Gastrointestinal symptoms were rarely reported as a key feature. We present a case report of a 74-year-old male who presented with symptoms of gastroenteritis and subsequently tested positive for COVID-19. This article aims to highlight an uncommon presentation of COVID-19 and that a high index of suspicion is required for COVID-19 in older people given their greater likelihood of presenting atypically.

## 1. Introduction

A cluster of viral pneumonia cases emerged in China in December 2019 secondary to a novel Coronavirus named severe acute respiratory syndrome coronavirus 2 (SARS-CoV-2) [1]. The clinical spectrum of disease appears wide and encompasses asymptomatic infection and mild upper respiratory tract illness to severe viral pneumonia leading to respiratory failure or even death [2]. The World Health Organisation named the resulting pneumonic disease coronavirus disease, COVID-19, and declared a pandemic on 12th March 2020. Published case series have listed fever (98%), cough (76%), myalgia/fatigue (44%) and dyspnea (55%) as common presenting symptoms in COVID-19 [2,3]. Diarrhoea (3–5%) was a rare presentation in the early case series [2,3], however clinicians have begun to question whether the prevalence of diarrhoea as a symptom of COVID-19 is underestimated [4]. Gathering and disseminating information on the range of clinical presentations of this novel viral illness is vital to enable prompt diagnosis, case isolation and treatment. Delays in diagnosis and case isolation in particular add to the public health challenge of COVID-19, enabling virus transmission. Therefore, we discuss a case of confirmed COVID-19 presenting with predominantly gastrointestinal symptoms in the United Kingdom (UK).

## 2. Case Presentation

Informed consent for this case report was given by the patient. He was a 74-year-old man with a past medical history of quiescent Ulcerative Colitis and Hypertension. He took Amlodipine 5 mg for Hypertension but took no regular medications for his Ulcerative Colitis and had never required immunomodulator therapy or surgery. The patient had not had a flare of his Ulcerative Colitis in over 15 years. He was independent with his activities of daily living and lived with his wife. The patient was first hospitalized in early March 2020 with a 3-day history of diarrhoea and vomiting following a trip 7 days prior to Scotland. No other family members were symptomatic. He complained of passing liquid brown stool and vomiting up to three times a day. At the point of admission, he had no further diarrhoea or vomiting episodes for the prior 24 h. On direct questioning he denied any respiratory symptoms and his chest X-ray revealed no radiological abnormalities. He remained apyrexial and his oxygen saturations were 96% on air. His blood results across his two admissions are summarized in Table 1 but of note he was hyponatremic at 122 mmol/L when he first presented to hospital. He was admitted to a side room and rehydrated with intravenous 0.9% Saline over 8 h and discharged within 24 h once his Sodium had risen >125 mmol/L with planned regular outpatient monitoring of his electrolytes.

The patient was re-admitted 72 h later with mild delirium and functional decline along with further intermittent loose stools and vomiting. The patient was spending most of the day in bed with symptoms of lightheadedness and fatigue. His appetite had decreased. The patient now reported a dry cough and described feeling feverish. The patient was admitted to a side room and a swab was sent for viral PCR. Forty-eight hours later, SARS-COV-2 viral RNA was detected and the patient was transferred to a COVID-19 cohort ward. The remainder of his respiratory viral screen was negative. He did not have any diarrhoea in hospital therefore no samples for norovirus, routine culture or Calprotectin were sent. Local guidelines do not advocate sending stool for SARS-CoV-2 RNA testing. Only 55% of stool samples are positive for SARS-CoV-2 RNA in patients with confirmed COVID-19 disease and there is no correlation between a detection of RNA in the stool and gastrointestinal symptoms [5]. Of note the patient never required oxygen therapy. Despite reporting feeling feverish, no objective fever was recorded during either admission. Furthermore, his lymphocyte count and platelet count were within the normal range on both admissions and his CRP peaked at 21 mg/L (Table 1). He was placed on a 1.5 L fluid restriction and his sodium normalized within 3 days. He was discharged after a 72-h inpatient stay and he has subsequently made a full recovery. The symptom constellation of diarrhoea and vomiting followed by the development of a dry cough in the context of a positive swab for SARS-CoV-2 led us to believe his entire presentation was COVID-19 disease.

## 3. Discussion

It is understandable that early case series from China focused on the respiratory symptoms associated with COVID-19, given that the most severe disease can cause respiratory failure and death. However, since they did not highlight diarrhoea or other gastrointestinal (GI) symptoms as presenting features of COVID-19, these symptoms may have been and continue to be under-recognized. As evidenced by our case report, this knowledge gap can lead to a delay in diagnosis and will increase the opportunity for viral transmission to other people including healthcare workers.

In contrast to those early reports, 17% of patients in a Singapore case series reported diarrhoea [6]. Additionally, more recent analysis of the clinical features of 204 patients presenting across Hubei province in China during February 2020 identified that one in five patients reported either vomiting, diarrhoea or abdominal pain at presentation [7]. However, most patients also reported respiratory symptoms and presented with a fever. Only six out of 204 patients presented with GI symptoms in the absence of respiratory symptoms and only one patient presented with GI symptoms without fever. Therefore, our case not only highlights the importance of recognizing GI symptoms in COVID-19 but also highlights the potential for cases to present with predominantly GI symptoms alone. Our case also suggests that GI symptoms are likely to be present in Western populations, where the disease has been present for a shorter time period and is therefore less comprehensively described at the time of writing.

It is postulated the SARS-CoV-2 binds to host ACE2 receptors on target cells to gain entry [8] and ACE2 is highly expressed in human small intestine [4]. ACE2 is a regulator of intestinal inflammation and can be associated with an increased risk of colitis [9]. Therefore, there is a plausible biological mechanism through which SARS-COV-2 could cause diarrhoea and other GI symptoms. Given the variable disease progression and time course in COVID-19, it is likely that over time the true aetiology of diarrhoea will prove multifactorial. The delayed focus on GI symptoms, compared with respiratory symptoms, is consistent with descriptions of other novel coronavirus diseases. For example, diarrhoea was subsequently recognized as a symptom of Middle East Respiratory Syndrome Coronavirus infection following initial descriptions which focused on respiratory complaints [10].

It has long been recognized within Geriatric Medicine that older people with any illness may present atypically. Some of the commonest atypical presentations of systemic illness are delirium, functional decline and falls [11,12] since acute illness leads to decompensation of pre-existing medical conditions. In our case, in addition to the gastrointestinal symptoms we saw a mild delirium, functional decline and a pattern of electrolyte disturbances consistent with Syndrome of Inappropriate Anti-Diuretic Hormone. It is currently unknown if older adults with COVID-19 present differently to younger patients and our patient was over 20 years older than the average age of the patient cohort from Hubei province [7]. To date, most triage of potential cases of COVID-19 has been based upon the presence of respiratory symptoms, however, given we know that older adults are less likely to develop fever or typical symptoms with other respiratory viral illnesses such as Influenza [13,14], there is an urgent need to properly describe the spectrum of clinical features in this patient group.

It is therefore imperative that as Geriatricians we use our expertise to maintain a high index of suspicion of COVID-19 in any older patient presenting to hospital as an emergency. High quality UK studies of older people are required to ascertain the prevalence of GI symptoms and any other non-respiratory presentations which may be consistent with COVID-19.

## Figures and Tables

**Table 1 geriatrics-05-00031-t001:** Results Summary.

Laboratory Characteristic	Admission 1	Admission 2	Discharge
White blood cell 10^9^/L	4	5.1	4.5
Haemoglobin 10^12^/L	152	138	137
Platelet count 10^9^/L	332	333	410
Lymphocyte count 10^9^/L	1.30	1.51	2.06
Sodium mmol/L	122	123	130
Potassium mmol/L	4	4.4	4.4
Creatinine μmol/L	49	52	58
Urine Sodium mmol/L	14.6	133.6	-
Urine Osmolality mOsm/Kg	547	596	-
Serum Osmolality mOsm/Kg	253	-	-
CRP mg/L	14	21	13
Lactate mmol/L	1.4	1	-
Bilirubin μmol/L	9.1	10.2	-
Alkaline Phosphatase U/L	99.9	101.2	-
Alanine Transferase U/L	22.7	27.7	-

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
