# Peer review of "COVID-19 and Gastrointestinal Symptoms—A Case Report"

_geriatrics, 2020, doi:10.3390/geriatrics5020031_

Round 1
Reviewer 1 Report
The authors present a 74 yo patient with PCR proven COVID-19. According to the authors the primary symptoms were diarrhea and vomiting. However, the patient also suffered from ulcerative colitis. How can the authors be sure that the patient´s symptoms were due to COVID-19 rather than due to UC? At least this point needs to be discussed.
What was the reason that stool was not evaluated for SARS-CoV-2?
It is well known that chest-x-ray lacks sensitivity for COVID-19 pneumonia. A negative chest x-ray does not exclude atypical pneumonia.
Diarrhea and vomiting has been described before as possible symptoms of COVID-19.
Author Response
Response to peer review comments
We appreciate the peer review of our case report and have considered the issues raised. Our response to the peer review comments are detailed below.
- The authors present a 74 yo patient with PCR proven COVID-19. According to the authors the primary symptoms were diarrhea and vomiting. However, the patient also suffered from ulcerative colitis. How can the authors be sure that the patient´s symptoms were due to COVID-19 rather than due to UC? At least this point needs to be discussed.
The patient was diagnosed with Ulcerative Colitis in 1976 and fortunately had been in long term remission. The patient had not had a flare of his Ulcerative Colitis for over 15 years. Additionally, the temporal association between the gastrointestinal symptoms and subsequent onset within a few days of a dry cough and feverishness strongly suggests all his symptoms were explained by COVID-19. This early experience has now been replicated in our trust with a number of other older patients also presenting with gastrointestinal pathology as their primary complaint, who have then gone on to develop more typical symptoms of COVID-19
The manuscript has been updated as follows:
Line 38 ‘past medical history of quiescent Ulcerative Colitis’
Line 40-41 ‘. The patient had not had a flare of his Ulcerative Colitis in over 15 years.’
Line 67-69 ‘The symptom constellation of diarrhoea and vomiting followed by the development of a dry cough in the context of a positive swab for SARS-CoV-2 led us to believe his entire presentation was COVID-19 disease’
- What was the reason that stool was not evaluated for SARS-CoV-2?
Stool samples for microscopy, culture, norovirus and faecal calprotectin were planned. However, the patient did not have further episodes of diarrhoea as an inpatient. Routine testing of stool samples for SARS-CoV-2 RNA is not part of the standard diagnostic protocol in the NHS. Furthermore, only 55% of patients with confirmed COVID-19 have positive stool samples of SARS-CoV-2 RNA and the presence of gastrointestinal symptoms is not associated with sample viral RNA positivity.
The manuscript has been updated as follows:
Lines 58-62 ‘He did not have any diarrhoea in hospital therefore no samples for norovirus, routine culture or Calprotectin were sent. Local guidelines do not advocate sending stool for SARS-CoV-2 RNA testing. Only 55% of stool samples are positive for SARS-CoV-2 RNA in patients with confirmed COVID-19 disease and there is no correlation between a detection of RNA in the stool and gastrointestinal symptoms [5].’
Updated reference: 5. Wu, Y. et al. Prolonged presence of SARS-CoV-2 viral RNA in faecal samples. The Lancet Gastroenterology & Hepatology 5, 434–435 (2020).
- It is well known that chest-x-ray lacks sensitivity for COVID-19 pneumonia. A negative chest x-ray does not exclude atypical pneumonia.
Our local data suggest chest x-ray abnormalities are present in 59% of cases and lung changes on CT scanning will demonstrate changes in 86% of patients. Not all patients will develop an atypical pneumonia and without significant respiratory symptoms further imaging is not warranted.
- Diarrhoea and vomiting has been described before as possible symptoms of COVID-19.
Internationally and locally there has been significant focus on the respiratory symptoms of COVID-19 potentially to the detriment of other recognised presentations. Most triage of potential COVID-19 disease has been based on the presence of respiratory symptoms. This case report was to highlight the principle that older people with any disease may present atypically and Geriatricians should apply the same principle in approaching COVID-19. This is an important message not only to improve patient care, but also to reduce the exposure of other patients and healthcare workers to COVID-19.
The manuscript has been updated as follows:
Lines 111-115 ‘This will inform triage and testing policies for potential COVID-19 cases which, to date, have mostly been based upon the presence of respiratory symptoms and fever. In the older adult population, such policies may lead to delay in diagnosis. Not only is such delay detrimental to patients but it could also lead to potentially avoidable exposure of other patients and healthcare workers to COVID-19.’
We hope that our revisions satisfy the concerns raised through the peer review process. If any further issues remain then please let us know and we would be happy to further revise the manuscript.
Reviewer 2 Report
This case report is interesting, well written and clear.
It addresses a relevant topic describing the gastrointestinal symptoms in a COVID-19 patient.
This topic has been addressed in some recently published papers; it is interesting to have another report on this specific and not so frequent clinical presentation of COVID-19
Author Response
This case report is interesting, well written and clear.
It addresses a relevant topic describing the gastrointestinal symptoms in a COVID-19 patient.
This topic has been addressed in some recently published papers; it is interesting to have another report on this specific and not so frequent clinical presentation of COVID-19
Thank you for your review and comments. We have updated our manuscript to answer comments from another reviewer.
Reviewer 3 Report
This is an excellently written case report of an important and common finding. We are also about to report similar findings in out symptom tracker that frail patients present particularly with GI symptoms and delirium. There is not much out there published on this at the moment and this well written case report is helpful.
Author Response
This is an excellently written case report of an important and common finding. We are also about to report similar findings in out symptom tracker that frail patients present particularly with GI symptoms and delirium. There is not much out there published on this at the moment and this well written case report is helpful.
Thank you for your review and comments. We have updated our manuscript to answer comments from another reviewer.
Round 2
Reviewer 1 Report
na